# Metabolite Profiling, Antioxidant and Key Enzymes Linked to Hyperglycemia Inhibitory Activities of *Satureja hispidula*: An Underexplored Species from Algeria

**DOI:** 10.3390/molecules27248657

**Published:** 2022-12-07

**Authors:** Ammar Haouat, Habiba Rechek, Diana C. G. A. Pinto, Susana M. Cardoso, Mónica S. G. A. Válega, Abdelhamid Boudjerda, Artur M. S. Silva, Ratiba Mekkiou

**Affiliations:** 1Unité de Valorisation des Ressources Naturelles, Molécules Bioactives et Analyse Physicochimiques et Biologiques (VARENBIOMOL), Université des Frères Mentouri, Constantine 25000, Algeria; 2Department of Chemistry, Faculty of Exact Sciences, University of Oued Souf, Oued Souf 39000, Algeria; 3LAQV-REQUIMTE & Department of Chemistry, University of Aveiro, 3810-193 Aveiro, Portugal; 4Department of Biology of Organisms, Faculty of Sciences of Nature and Life, University of Batna 2, Mostefa Ben Boulaid, Batna 05078, Algeria; 5Laboratory of Cellular and Molecular Biology, University of Mohammed Seddik Benyahia, Jijel 18000, Algeria

**Keywords:** *Satureja hispidula*, UHPLC-DAD-ESI/MS analysis, antioxidant activity, α-glucosidase, α-amylase

## Abstract

In the present study, two extracts from the aerial parts of the endemic species *Satureja hispidula* were analyzed for the first time by ultra-high-performance liquid chromatography coupled with a diode array detector and an electrospray mass spectrometer (UHPLC-DAD-ESI/MS) method in order to identify and quantify their phenolic compounds. These extracts’ antioxidant, α-glucosidase and α-amylase inhibitory activities were also evaluated. UHPLC-DAD-ESI/MS allowed the identification of 28 and 20 compounds in the ethanolic and aqueous extracts, respectively; among them, 5-*O*-caffeoylquinic acid was the most abundant in both extracts. The biological assay results indicate that the species *S. hispidula*, besides its high antioxidant power, is also potentially useful for inhibiting the α-glucosidase enzyme. In both antioxidant and α-glucosidase inhibitory assays, the aqueous extract exhibited the most promising results, significantly better than the standards used as positive controls.

## 1. Introduction

Diabetes mellitus is a metabolic disorder characterized by high blood glucose levels and insufficient insulin production by the pancreas. Numerous studies have shown that free radicals and other oxidants have been substantially recognized to play a significant role in the development and complication of certain pathological disorders, including diabetes mellitus [1]. On the other hand, phenolic compounds have strong antioxidant properties and demonstrate promising potential in managing type 2 diabetes. Several antidiabetic medications are in use, but more efficient and safer agents are still needed. This explains the extensive research to find new active agents, particularly those that block important enzymes such as α-glucosidase and α-amylase.

Some compounds derived from plants exhibit this function, and the Algerian flora is a crucial source for researchers looking for new molecules. Among the Algerian species is *Satureja hispidula* Briq. (Lamiaceae), traditionally named summer savory and considered a medicinal plant for which several uses were reported [2], although not scientifically confirmed. Although its use in traditional Algerian medicine is not reported [3,4], there is information concerning *S. hispidula* aerial parts used in Turkey and Iran to treat several illnesses, including hyperglycemia [5]. Interesting biological properties were reported, mainly for *S. hispidula* essential oil [6], but considering these biological properties and the ones also established for other *Satureja* species [5,7]. *S. hispidula* is still considered an underexplored species. Moreover, its antihyperglycemic activity needs to be established. Thus, this study was conducted to assess for the first time the phytochemical composition as well as the antioxidant and α-glucosidase and α-amylase inhibitory properties of the aqueous and ethanolic extracts from *Satureja hispidula* growing in Algeria, for which no prior phytochemical and biological data have been described.

## 2. Results

### 2.1. Total Bioactive Content

Table 1 shows the values obtained for the total content of phenolic compounds and flavonoids. According to the findings, the plant contains significant amounts of both phenolic and flavonoid contents.

The total phenolic content (TPC) of *S. hispidula* aqueous extract was reported to be 131.24 ± 0.81 mg GAE/g extract. TPC levels were substantially greater than in the ethanol extract (85.7 ± 0.46 GAE/g extract). However, in terms of total flavonoid content (TFC), no significant difference was found between the ethanol extract (22.9 ± 0.21 mg QE/g of extract) and the aqueous (22.5 ± 1.55 mg QE/g extract) extract.

### 2.2. UHPLC-DAD-ESI/MS Characterization of S. hispidula Extracts

The phenolic content of *S. Hispidula* ethanolic and aqueous extracts was analyzed using the UHPLC-ESI-MS/MS technique (Figure 1). As stated in Table 2, a total of 27 and 17 phenolic compounds were detected for the first time in the ethanol and aqueous extracts, respectively. Overall, both chromatograms were characterized by the presence of several phenolic acids and flavonoids tentatively identified based on their fragmentation patterns and the data reported in Table 2. Even though UHPLC-ESI-MS/MS analysis of the ethanolic and aqueous extracts of *S. hispidula* revealed that their phenolic composition was quite different, the major compound in both cases was 5-*O*-caffeoylquinic acid (5-*O*-CQA) (Figure 1; Table 2).

Seven phenolic acid derivatives (peaks **1**, **3**, **5**, **6**, **9**, **17**, and **21**) were identified in both extracts. Based on their fragmentation patterns, compared with standards injected in the same conditions and data in the literature [8,9,10], these compounds were assigned to 3-*O*-CQA, 5-*O*-CQA, 4-*O*-CQA, caffeic acid, 1-*O*-CQA, 4,5-*O*-CQA and rosmarinic acid. Except for caffeic acid, the other derivatives were found in high quantities, especially in the aqueous extract (Table 2).

Thirteen flavonoid derivatives (peaks **8**, **10–14**, **16**, **19**, **20**, **22**, **24**, **25**, **29**) were found in the ethanolic extract, and 10 in the aqueous extract (peaks **8**, **11–15**, **19**, **20**, **23**, **24**). The most abundant were quercetin and apigenin derivatives (Table 2), although, in the ethanolic extract, the most significant compound was ophiopogonanone E (peak **24**) (Figure 1b), which could not be quantified because we did not have the proper standard, and was identified based on the fragmentation detected [11,12,13]. The remaining compounds were identified by comparing our data with those reported in the literature [14,15,16,17,18,19,20,21,22,23,24], although in some cases, a commercial standard was injected in the same chromatographic conditions. In the ethanolic extract, the flavonoid derivative found in higher amounts was kaempferol-*O*-glucoside (peak **18**), whereas in the aqueous extract, it was apigenin 7-*O*-glucuronide (peak **20**) (Table 2).

Other compounds were also present in the ethanolic extract, albeit in lower concentrations, namely oxodihydroxy octadecenoic acid, trihydroxyoctadecenoic acid, and dihydroxyoctadecanoic acid (Table 2).

Although the extracts were quite different, several compounds occurred in both extracts, but in different amounts (Table 2). It should be noted that, compared to the ethanolic extract, the aqueous extract was particularly rich in caffeoylquinic acids, occurring at 67.13 and 27.57 µg/mg of extract, respectively. Furthermore, the aqueous extract also showed higher flavonoid glycoside content (46.92 µg/mg of extract) than the ethanolic extract (30.28 µg/mg of extract).

### 2.3. Antioxidant Activity

The antioxidant potential of both ethanolic and aqueous extracts was estimated through three different antioxidant assays (DPPH^●^, ABTS^●+^ and FRAP). The results are shown in Table 3.

The aqueous extract showed higher DPPH and ABTS radical scavenging activities compared to the ethanolic extract. The IC_50_ of DPPH and ABTS of the aqueous extract were 2.43 ± 0.11 and 2.16 ± 0.1 μg/mL, respectively, and were more than 1.5 times greater than that of the standard ascorbic acid used as control (IC_50_ = 3.86 ± 0.23 and 4.01 ± 0.09 μg/mL in DPPH and ABTS assays, respectively). Although its activity was lower than that of the aqueous extract, the ethanolic extract showed very good antioxidant activity, as revealed by the DPPH and ABTS tests with IC_50_ values of 5.67 ± 0.07 and 5.95 ± 0.82 μg/mL, respectively. On the other hand, the same tendency was also observed in the FRAP assay, in which the aqueous extract (A_0.5_ = 5.82 ± 0.17 μg/mL) displayed better activity compared to the ethanolic extract (A_0.5_ = 10.77 ± 0.12 μg/mL) and the standard ascorbic acid (A_0.5_ = 10.03 ± 0.22 μg/mL).

### 2.4. Inhibition of Enzymatic Activities

The key metabolic enzymes linked to hyperglycemia are commonly employed to assess a substance’s antidiabetic potential. In the present study, the ability of the ethanolic and aqueous extracts of *S. hispidula* to inhibit the α-glucosidase and α-amylase enzymes was assessed. The results are shown in Table 4.

The statistical analysis performed for the results obtained in the α-glucosidase inhibition assay method noted significant differences (with *p* < 0.05) in the inhibitory activities of the aqueous and ethanol extracts of *S. hispidula* against this enzyme. The aqueous extract displayed the highest inhibitory activity against the α-glucosidase enzyme, with IC_50_ values in mg/mL of 23.52 ± 6.33 μg/mL when compared to the ethanolic extract showing IC_50_ = 106.94 ± 1.55 μg/mL. Still, both extracts proved to have more potent α-glucosidase inhibitory activity than the positive control, Acarbose (IC_50_ = 405.77 ± 34.83 μg/mL). However, with regard to α-amylase inhibitory activity, the aqueous extract (4.86 ± 0.004%) was shown to be less potent than the ethanol extract (30.34 ± 4.58%) in inhibiting the enzymatic activity of α-amylase tested at the concentration of extract equal to 10 mg/mL (Table 4).

## 3. Discussion

### 3.1. Compound Identification

The caffeoylquinic acid derivatives presented the expected fragmentation pattern. The monocaffeoyl showed a pseudomolecular ion [M-H]^−^ at *m/z* 353, and the dicaffeoyl a pseudomolecular ion [M-H]^−^ ion at *m/z* 515. Typically, in dicaffeoylquinic acids, the fragment ion at *m/z* 353 is observed, which is produced by the loss of a caffeoyl moiety [8]. The remaining fragments are similar to the ones produced by the monocaffeoylquinic acid derivatives (Table 2), which comprise the fragments at *m/z* 191 and *m/z* 179, confirming the presence of the quinic and caffeic acid moieties, respectively [8,9,10]. Due to the loss of water or carbon dioxide, other interesting fragments could also be detected, as in the case of 5-*O*-CQA (Figure 2), the major metabolite found in the plant extracts (Table 2).

In the case of rosmarinic acid, with a pseudomolecular ion [M-H]^−^ at *m/z* 359, the fragmentation was quite different because the caffeoyl moiety is not linked to a quinic moiety but is instead a dimer of two caffeoyl moieties. Naturally, the fragment at *m/z* 179, confirming the presence of the caffeic acid moiety, was detected, as well as the fragment at *m/z* 181, confirming the saturated form (Table 2). Other essential fragments were the *m/z* 225 and *m/z* 197 (Figure 3), as well as the injection of a pure standard in the same chromatographic conditions to ensure the proposed assignment. Peak **8** showed an [M-H]^−^ ion at *m/z* 609 and was assigned to kaempferol-*O*-diglucoside. This ion suffered successive fragmentations by losing two glucosyl moieties, producing a peak at *m/z* 447 and another peak at *m/z* 289 typical of kaempferol glucosides. Peak **10** was assigned to quercetin 3-*O*-(6′′-malonylglucoside)-7-*O*-glucoside based on data in the literature and its MS^2^ pattern displaying [M-H]^−^ at *m/z* 667 and releasing MS^2^ fragment ions at *m/z* 505, 463, 301 [15]. Peak **11** displayed pseudomolecular ion [M-H]^−^ at *m/z* 593 and as prominent peaks at *m/z* 473 [M-H-120]^−^, *m/z* 431, and *m/z* 268, suggesting that this compound is apigenin-6,8-di-C-glucoside, according to MS data in the literature [16].

Peak **12** was assigned to kaempferol 3-*O*-(6"-acetylglucoside)-7-O-glucoside based on its pseudomolecular ion [M-H]^−^ occurring at *m/z* 651 and releasing its main fragment at *m/z* 489, which corresponded to kaempferol-(6"-glucoside), and another typical fragment at *m/z* 285 corresponding to kaempferol [17]. Peak **13**, with [M-H]^−^ at *m/z* 609 and generated fragments at *m/z* 343 and 301, was assigned to rutin [18], and peak **14** presented a pseudomolecular ion [M-H]^−^ at *m/z* 463, releasing an MS^2^ fragment at *m/z* 301 ([M-162]^−^, loss of a glucose moiety), corresponding to quercetin, which allowed the identification of the compound as quercetin-glucoside [17,19]. Peak **15** displayed ([M-H]^−^ at *m/z* 461, and the main MS^2^ fragment at *m/z* 285 ([M-H-176]^−^ due to the loss of glucuronyl moiety was tentatively identified as luteolin-7-O-glucuronide [20]. Peak **16** showing pseudomolecular ion at *m/z* 447 was identified as luteolin-7-O-glucoside by comparing its retention time and MS pattern with the commercial standard injected in the same chromatographic conditions. Peak 18 yielded [M-H]^−^ at *m/z* 447 and the main MS^2^ fragment at *m/z* 285 [M-H-162]^−^ resulting from the loss of one hexosyl moiety. Thus, based on its fragmentation pattern and UV spectrum, this compound was tentatively assigned to kaempferol-O-glucoside [19]. Peak 19 showing [M-H]^−^ at *m/z* 505 and MS^2^ fragments at *m/z* 463 [M-H-42]^−^ and 301 [M-H-162+42]^−^ due to the loss of acetyl and acetyl-O-glucoside moieties, respectively, was assigned to quercetin-O-acetyl glucoside. Peak 20 showed [M-H]^−^ at *m/z* 445 and the main fragment ion at *m/z* 269 due to the loss of a glucuronyl unit, so it was assigned to apigenin-7-*O*-glucuronide [21]. Peak 22 with [M-H]^−^ at *m/z* 489 and MS^2^ fragments at *m/z* 327 [M-H-120]^−^ and 285 [M-H-162+42]^−^, due to the loss of acetyl and acetyl-*O*-glucoside moieties, respectively, was tentatively assigned to luteolin-*O*-acetyl glucoside. Peak 23 exhibited the [M-H]^−^ ion at *m/z* 537, producing the main fragment at *m/z* 519 [M-H-H_2_O]^−^, and another fragment ion at *m/z* 269 allowed us to assign it to C-C linked biapigenin (2,8’’-biapigenin) [22]. Peak 25 was tentatively identified as dimethylquercetin based on its fragmentation pattern exhibiting [M-H]^−^ ion at *m/z* 329 and fragment ions at *m/z* 314 ([M-H-15]^−^) and *m/z* 299 ([M-H-15+15]^−^) resulting from successive loss of one and two methyl groups, respectively [23]. Peak 29 displayed a pseudomolecular ion [M-H]^−^ at *m/z* 343, producing the main fragment at *m/z* 328 [M-H-15+15+15]^−^ resulting from the loss of three methyl groups from the ion at *m/z* 343. Given that and based on data in the literature, this compound was tentatively assigned to 5,2’-dihydroxy-7,8,6’-trimethoxyflavone [24].

Finally, peaks **24** and **26** exhibited similar fragmentation patterns and a pseudomolecular ion at *m/z* 359, producing the main fragment at *m/z* 344 [M-H-15]^−^. Based on a literature search and the possible fragmentation (Figure 3), these compounds were identified as the flavonoid ophiopogonanone E and an isomer [11,12,13].

### 3.2. Biological Activities

The term “oxidative stress” refers to an attack on cells by free radicals, also known as “reactive oxygen species” (ROS). The accumulation of ROS in the body results in significant tissue damage. Many illnesses, most notably cancer, diabetes, and neurological disorders, are known to be linked to oxidative stress. When ROS accumulate in the cell, they can be neutralized by antioxidant molecules. There are two types of antioxidants: natural and synthetic. The latter has been linked to many harmful side effects. Thus, the search for non-toxic, natural antioxidant compounds has increased recently. Plants are an essential source of natural antioxidants due to the secondary metabolites they contain. These plants can be screened by assessing their antioxidant activity using various in vitro assays. Because numerous processes are involved in neutralizing free radicals, using one single test makes it impossible to determine the real antioxidant potential [25]. As a result, the *S. hispidula* extracts herein studied were analyzed through three different tests (DPPH, ABTS, and FRAP) to further measure the overall antioxidant potential.

The results for the antioxidant activity of both aqueous and ethanolic extracts of *S. hispidula* species could be related to their high content of hydroxycinnamic acids, since it was previously reported that such substances displayed strong antioxidant activities [26,27,28]. It was not possible to compare the obtained results to the literature. Indeed, we found no information on the biological activity of *S. hispidula* aerial parts in the literature. This work is, therefore, the first contribution to the study of the biological potential of this plant.

Diabetes mellitus is a chronic non-transmissible metabolic disorder characterized by hyperglycemia. Diabetes affects 422 million people across the world [29]. This number is continually increasing, with 642 million people predicted to be affected by the disease in 2040 [30]. Inhibiting the activity of digestive enzymes such as α-glucosidase and α-amylase, which limit the absorption of glucose derived from starch, is among the current strategies standing out for controlling hyperglycemia [31]. Many diabetes medications, such as acarbose, voglibose, and miglitol, act as enzymatic antagonists of α-glucosidase, reducing intestinal glucose absorption and postprandial glycemia. However, these inhibitors produce harmful side effects such as flatulence, diarrhea, and abdominal pain [31]. Thus, the search for new compounds with antidiabetic potential has increased recently. Medicinal plants with little toxicity and few side effects are critical therapeutic possibilities for treating diabetes [32]. The therapeutic effects of these plants have been linked to the presence of phytochemicals such as alkaloids, terpenoids, flavonoids, carotenoids, and β-glycans. These bioactive constituents provide therapeutic effects by combating reactive oxygen species or acting as hypoglycemics.

The aqueous extract had shown the highest content of rosmarinic acid (Table 2), which suggests that this compound may play an essential role in the inhibitory activity against the α-glucosidase enzyme. Hence, rosmarinic acid isolated from the leaf extract from *Portulaca oleracea* species exhibited a strong α-glucosidase inhibitory effect [33]. The flavonoid glycosides, highly present in ethanolic and aqueous extracts, may also contribute to their strong α-glucosidase inhibitory activity herein recorded. In the present work, both aqueous and ethanol extracts showed more potent inhibition of α-glucosidase enzyme than α-amylase. This behavior is beneficial because it is known that a strong inhibition of the α-glucosidase enzyme and a weak inhibition of the α-amylase enzyme has the advantage of having fewer adverse effects compared to those caused by an excessive inhibition of the α-amylase enzyme, as this results in an abnormal bacterial fermentation of undigested carbohydrates in the colon and increases abdominal discomfort [34,35].

The dual antioxidant and α-glucosidase inhibitor potential recorded for the two *S. hispidula* extracts suggest that this plant could be used as a natural remedy to control type 2 diabetes or as an excellent source of new molecules for the development of antidiabetic drugs. Indeed, several experimental and clinical studies have shown that diabetes is associated with a high level of oxidative stress [36,37,38]. Some of the processes involved in oxidative stress in diabetic patients include oxygen free radicals created by glycosylation reactions and changes in the content of the endogenous antioxidant defense systems [39]. On the other hand, hyperglycemia could also be the consequence of a significant increase in oxidative stress.

In such a context, it appears important to control both hyperglycemia and oxidative stress in diabetic patients by using medications that jointly target these two pathological disorders.

## 4. Materials and Methods

### 4.1. Chemicals

Ascorbic acid, 1,1-diphenyl-2-picrylhydrazyl (DPPH^●^), 2,2′-azinobis(3-ethylbenzothiazoline-6-sulfonic acid) diammonium salt (ABTS^●+^), neocuproine, trichloroacetic acid (TCA), and potassium ferricyanide were used for the determination of the antioxidant activities and were purchased from Sigma Chemical Co. (Sigma-Aldrich GmbH, Stern-heim, Germany). The following chemicals were used for the enzyme inhibition assays: α-glucosidase from *Saccharomyces cerevisiae*, 4-nitrophenyl α-D-glucopyranoside (pNPG), α-amylase from porcine pancreas, β-nicotinamide adenine dinucleotide (β-NADH), phenazinemethosulphate (PMS), nitrotetrazolium blue chloride (NBT) and ascorbic acid were obtained from Sigma (St. Louis, MO, USA), while acarbose and potato starch were purchased from Fluka (Bucharest, Romania) and Fisher (Pittsburgh, PA, USA), respectively. Sodium nitroprusside, sulfanilamide, and 3,5-dinitrosalicylic acid (DNS) were obtained from Acros Organics (Hampton, NH, USA). Folin–Ciocalteu reagent, Na_2_CO_3_, and gallic acid were purchased from Panreac (Barcelona, Spain). Standards used for the elucidation of the phenolic compounds and for elaboration of the calibration curves were obtained from EXTRA SYNTHESE (Genay Cedex, France). Acetonitrile HPLC-grade and formic acid were purchased from Panreac (Barcelona, Spain). All other chemicals were of analytical grade.

### 4.2. Extract Preparation

Aerial parts of *Satureja hispidula* Briq. were collected from Jijel province (northeastern Algeria, 36°40′36″ N, 5°57′09″ E) in March 2019 and authenticated by Prof. Boudjerda Azeddine (University of Jijel, Algeria). The leaves were dried at room temperature (25 °C) and then ground into powder.

Separate extractions were performed using two solvents: water and ethanol. To prepare the aqueous extract, 10 g of the plant powder was mixed with 200 mL of distilled water and boiled for 15 min before filtering. The resultant solution was frozen, freeze-dried, and kept at 4 °C until needed. In turn, an ethanol extract was prepared by macerating 10 g of powder in 200 mL of ethanol at room temperature for 48 h, followed by solvent replacement for two additional times. The ethanol was then removed at reduced pressure.

### 4.3. Total Bioactive Content

The Folin–Ciocalteu method, previously described by [40], was used to determine the total phenolic content (TPC) [40]. In brief, 15 µL from each plant extract (1 mg/mL) were combined with 15 µL of Folin’s reagent and 60 µL of water. After incubating the mixture for 5 min, 150 µL of Na_2_CO_3_ solution (20% *w*/*v*) was added. The mixture was incubated in the dark for another 60 min, and absorbance at 760 nm was measured. The total phenolic content was reported as mg gallic acid equivalents per gram of extract (mg GAE g of extract).

The total flavonoid content (TFC) was assessed using a slightly modified technique reported by [41]. In brief, 100 µL of each plant extract was combined with 100 µL of a 2% solution of aluminum chloride (AlCl_3_). After incubating the mixture at room temperature for 10 min, the absorbance at 415 nm was measured. The total phenolic content was reported as mg of quercetin equivalents per gram of extract (mg QE/g of extract).

### 4.4. UHPLC-DAD-ESI/MS Characterization of S. hispidula Extracts

The phenolic composition of *S. hispidula* ethanolic and aqueous extracts was determined using an analytical liquid chromatograph, the Dionex Ultimate 3000 (Dionex Co., San Jose, CA, USA) apparatus equipped with a diode array detector (Dionex Co. San Jose, CA, USA) coupled to a Thermo LTQ XL mass spectrometer (Thermo Scientific, San Jose, CA, USA) with an electrospray ionization source operating in negative mode. The column used was a Hypersil Gold (Thermo Scientific, USA) C18 column (100 mm length; 2.1 mm i.d.; 1.9 μm particle diameter, end-capped). The eluents used were a mixture of formic acid [0.1%(*v*/*v*)] in water (A) and acetonitrile (B) with an injection volume of 10 µL and a flow rate of 0.2 mL/min. The elution gradient was 5% (solvent A) for 14 min, 40% (solvent A) over 2 min, 100% (solvent A) over 7 min, and the re-equilibration of the column with 5% of solvent A for 10 min. The complete scan encompassed the mass range from *m*/*z* 100 to 2000. UV-Vis spectral data were collected between 200 and 500 nm, while chromatogram profiles were obtained at 280 nm. A Thermo Xcalibur Qual Browser data system (Thermo Scientific, USA) was used to analyze the spectra, and chemicals were tentatively identified by comparing their mass spectra and retention durations to what had previously been described in the literature.

Stock solutions of standard references were prepared in the same conditions. Quantification was carried out using the external standard method. The identification of peaks was confirmed by comparing their retention times and MS and DAD spectra with those of reference, or by comparing with MS and UV-Vis data previously reported in the literature.

### 4.5. Antioxidant Activities

#### 4.5.1. Determination of 1,1-Diphenyl-2-Picrylhydrazyl Radical Scavenging Activity

The antioxidant activity was tested by inhibiting the free 1,1-diphenyl-2-picrylhydrazyl radical (DPPH^●^), as described by [42]. In brief, 250 µL of DPPH solution (8.66 × 10^−5^ M) were mixed with 50 µL of each extract concentration. The mixtures were incubated at room temperature in the dark for 30 min. The absorbance was then observed at 734 nm. The negative control was prepared in the same way, but with the extract replaced by its solvent. The following formula was used to measure the percentage of inhibition:DPPH Scavenging effect = (Ac − As)/Ac 100%
where the absorbance of the control is represented by Ac, and the absorbance of the sample is represented by As.

The sample concentration required to inhibit 50% of the reaction (IC_50_) was measured. The results were then compared to the ascorbic acid used as reference standard.

#### 4.5.2. Determination of 2,2′-Azinobis(3-Ethylbenzothiazoline-6-Sulfonic Acid) Scavenging Activity

The *S. hispidula* extracts’ free radical scavenging ability was assessed using an ABTS^●^^+^ decolorization test developed by [43]. In brief, 250 µL of ABTS^•+^ were mixed with 50 µL of each extract concentration. The mixture was incubated at room temperature in the dark for 20 min before the absorbance at 734 nm was measured. The following equation was used to determine ABTS^+^ scavenging ability:ABTS^•+^ Scavenging effect = (Ac − As)/Ac 100%
where Ac represents the absorbance of the control, and As represents the absorbance of the sample. The sample concentration capable of inhibiting 50% of the reaction (IC_50_) was measured. The results were then compared to the ascorbic acid used as reference standard.

#### 4.5.3. Ferric Reducing Antioxidant Power Assay

Total antioxidant potential of *S. hispidula* was measured by the ferric reducing antioxidant power assay (FRAP) according to the technique described by [44]. In a redox-linked colorimetric process, the colorless ferric iron (Fe^3+)^ is reduced to bright blue ferrous iron (Fe^2+^) using antioxidants present in the samples as reductants. A 200 µL amount of sample/standard and 200 µL of potassium ferrocyanide (1% *w*/*v*) were mixed concurrently. After 20 min at 50 °C, 200 µL of TCA (10% *w*/*v*) were added. The mixture was then transferred to a 96-well plate, and 75 µL of deionized water and 15 µL of iron(III) chloride (0.1% *w*/*v*) were added at the same time. Following ferric iron reduction, a blue color appeared that could be measured colorimetrically at 690 nm. The results were then compared to the ascorbic acid used as reference standard.

### 4.6. Inhibition of Enzymatic Activities

#### 4.6.1. Inhibition of α-Glucosidase Activity

The α-glycosidase test was carried out using a slightly modified version of the technique reported by [45]. The *p*-nitrophenyl glycopyranoside (pNPG) substrate solution was produced in 20 mM phosphate buffer, pH 6.9. The extract (50 µL) was combined with a 50 µL solution of 4-nitrophenyl-D-glucopyranoside (PNPG). The reaction was then started by adding 100 µL of the α-glucosidase enzyme solution to the mixture. At 37 °C, the absorbance was measured at 405 nm every minute for 20 min. The negative control was prepared in the same way, but with the extract replaced by its solvent.

The following formula was used to compute the % inhibition:(Ac − As)/Ac × 100 = Inhibition%
where Ac denotes the absorbance of the control (enzyme and buffer), and As denotes the absorbance of the sample (enzyme and buffer).

#### 4.6.2. Inhibition of α-Amylase Activity

The α-amylase inhibition assay was carried out with minor changes to the technique previously published by [45]. A 200 µL amount of each extract concentration was combined with 400 µL of a 0.8% (*w*/*v*) starch solution on a 96-well plate and incubated for 5 min at 37 °C. The reaction was initiated by the addition of 200 µL of α-amylase. After that, 200 µL of the mixture was combined with 600 mL of 3,5-dinitrosalicylic acid (DNS) reagent. A second aliquot of 200 µL of the first mixture was taken and mixed with DNS reagent around 15 min later. Then, 250 µL of each combination was heated at 100 °C for 10 min before being transferred to a 96-well microplate, and the absorbance at 450 nm was measured. The inhibiting action against α-amylase was estimated as inhibition percentage using the following formula:(Ac − As)/Ac × 100 = Inhibition%
where Ac represents the absorbance of the control (enzyme and buffer), and As represents the absorbance of the sample (enzyme and inhibitor).

The 50% inhibitory concentration (IC_50_) of a sample was measured. The results were then compared to acarbose, a commercial standard.

### 4.7. Statistical Analysis

All calculations were performed in MINITAB software (version 16, State College, PA, USA). Data from three separate assays were reported as the mean standard deviation (m SD). The data recorded as the mean ± standard deviation (m ± SD) were submitted to analysis of variance (ANOVA), followed by a post hoc honestly significant difference (HSD) Tukey’s test at *p* < 0.05.

## 5. Conclusions

The present study is the first report devoted to the identification and quantification of the phytochemical constituents of aerial parts of the endemic species *S. hispidula* growing in Algeria, as well as the assessment of their antioxidant and antidiabetic effects. The results indicate that aqueous and ethanolic extracts exhibited high antioxidant and α-glucosidase inhibitory activities; the aqueous extract was the most efficient. Finally, the current findings support the use of *S. hispidula* in Algerian folk medicine to treat hyperglycemia, suggesting it is an essential source for developing new antidiabetic medications.

## Figures and Tables

**Figure 1 molecules-27-08657-f001:**
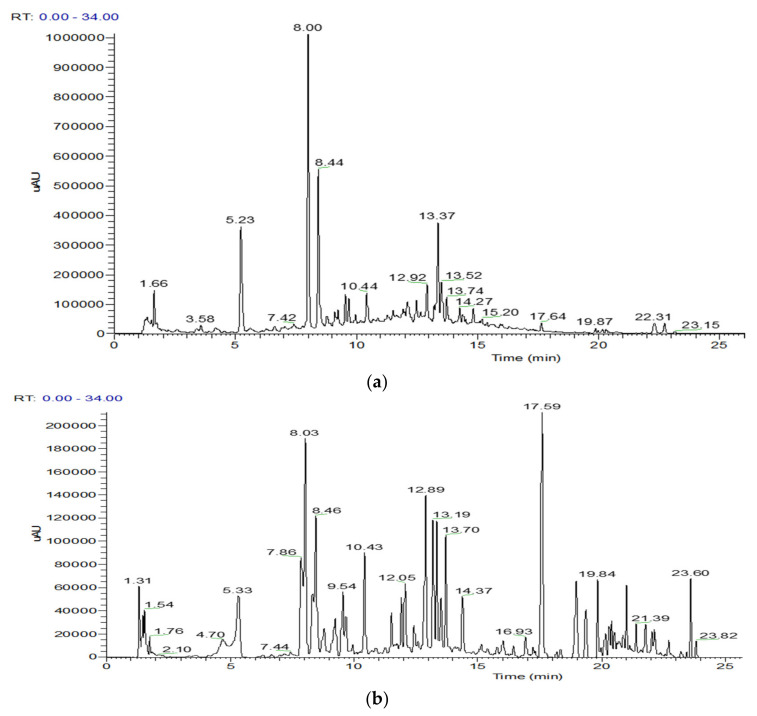
UHPLC chromatogram of the aqueous (**a**) and ethanolic (**b**) extracts of *S. hispidula* recorded at 280 nm.

**Figure 2 molecules-27-08657-f002:**
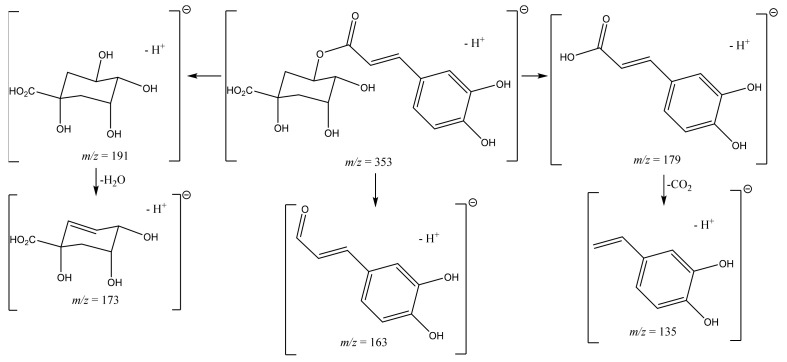
5-*O*-CQA main fragments.

**Figure 3 molecules-27-08657-f003:**
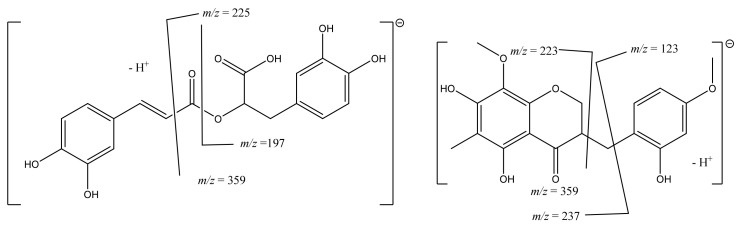
Rosmarinic acid and ophiopogonanone E main fragments.

**Table 1 molecules-27-08657-t001:** Total phenolic and flavonoid contents of *S. hispidula* extracts.

Sample	TPC (mg GAE/g of Extract)	TFC (mg QE/g of Extract)
*S. hispidula* EE	85.7 ± 0.46	22.9 ± 0.21
*S. hispidula* AE	131.24 ± 0.81	22.5 ± 1.55

TPC: total phenolic compounds, TFC: total flavonoid compounds, GAE: gallic acid equivalent, QE: quercetin equivalent, EE: ethanolic extract; AE: aqueous extract.

**Table 2 molecules-27-08657-t002:** Phenolic composition of *S. hispidula* ethanolic and aqueous extracts by UHPLC-DAD-ESI/MS.

N°	Rt (min)	[M-H]^−^	MS^2^ Fragments	Assigned Identification	Quantification (µg/mg of Extract)	
					EE	AE
**1**	5.23 5.33	353	191 [M-H-caffeoyl]^−^ 179 [M-H-quinic]^−^ 173 [M-H-caffeoyl-H_2_O]^−^ 135 [M-H-quinic-CO_2_]^−^	3-*O*-CQA	4.54 ± 0.49	13.39 ± 2.39
**2**	7.86	353	191 [M-H-caffeoyl]^−^ 179 [M-H-quinic]^−^ 163 [M-H-*O*-quinic]^−^ 135 [M-H-quinic-CO_2_]^−^	5-*O*-CQA *	3.02 ± 0.00	/
**3**	8.00 8.03	353	191 [M-H-caffeoyl]^−^ 179 [M-H-quinic]^−^ 173 [M-H-caffeoyl-H_2_O]^−^ 163 [M-H-*O*-quinic]^−^ 135 [M-H-quinic-CO_2_]^−^	5-*O*-CQA	9.85 ± 2.04	26.05 ± 4.89
**4**	8.33	353	191 [M-H-caffeoyl]^−^ 179 [M-H-quinic]^−^ 163 [M-H-*O*-quinic]^−^	4-*O*-CQA *	1.46 ± 0.60	/
**5**	8.44 8.46	353	191 [M-H-caffeoyl]^−^ 179 [M-H-quinic]^−^ 173 [M-H-caffeoyl-H_2_O]^−^ 163 [M-H-*O*-quinic]^−^	4-*O*-CQA	6.32 ± 1.14	13.97 ± 2.52
**6**	8.85	179	135 [M-H-CO_2_]^−^	Caffeic acid	0.55 ± 0.04	NQ
**7**	9.12	225	97	Unknown	NQ	NQ
**8**	9.34	609	447 [M-H-glucosyl]^−^ 285 [M-H-diglucosyl]^−^	Kaempferol-*O*-diglucoside	0.69 ± 0.23	1.91 ± 0.65
**9**	9.58	353	191 [M-H-caffeoyl]^−^ 179 [M-H-quinic]^−^ 173 [M-H-caffeoyl-H_2_O]^−^ 135 [M-H-quinic-CO_2_]^−^	1-*O*-CQA	NQ	NQ
**10**	9.64	667	505 [M-H-glucosyl]^−^ 463 [M-H-6″-malonylglucoside]^−^ 301 [quercetin moiety]^−^	Quercetin 3-*O*-(6″-malonylglucoside)-7-*O*-glucoside	2.84 ± 0.28	/
**11**	9.77	593	575 [M-H-H_2_O]^−^ 503 [M-H-C_3_H_6_O_3_]^−^ 473 [M-H-C_4_H_8_O_4_]^−^ 431 [M-H-glucosyl]^−^ 268 [apigenin moiety]^−^	Apigenin 6,8 di-*C*-glucoside	1.59 ± 0.17	4.44 ± 0.78
**12**	10.43 10.44	651	531 [M-H-C_4_H_8_O_4_]^−^ 489 [M-H-glucosyl]^−^ 447 [M-H-glucosyl-CH_3_CO]^−^ 285 [kaempferol moiety]^−^	Kaempferol 3-*O*-(6”-acetylglucoside)-7-*O*-glucoside	4.05 ± 0.40	5.11 ± 1.57
**13**	11.63	609	462 [M-H-rhamnosyl]^−^ 342 [M-H-rhamnosyl-C_4_H_8_O_4_]^−^ 301 [quercetin moiety]^−^	Rutin	0.52 ± 0.13	2.21 ± 0.80
**14**	12.03	463	301 [quercetin moiety]^−^	Quercetin-glucoside	1.07 ± 0.20	3.38 ± 0.93
**15**	12.15	461	285 [luteolin moiety]^−^	Luteolin-7-*O*-glucuronide	/	6.27 ± 0.75
**16**	12.19	447	327 [M-H-C_3_H_6_O_3_]^−^ 285 [luteolin moiety]^−^ 178 ^0,4^B^−^ 151 ^1,3^A^−^	Luteolin-7-*O*-glucoside	2.12 ± 0.30	/
**17**	12.89 12.92	515	353 [M-H-caffeoyl]^−^ 335 [M-H-caffeoyl-H_2_O]^−^ 309 [M-H-caffeoyl-CO_2_]^−^ 191 [quinic moiety]^−^ 179 [caffeoyl moiety]^−^	4,5-*O*-diCQA	7.43 ± 0.39	4.98 ± 0.05
**18**	13.19	447	285 [kaempferol moiety]^−^	Kaempferol-*O*-glucoside	9.06 ± 0.79	/
**19**	13.34 13.37	505	462 [M-H-CH_3_CO]^−^ 301 [quercetin moiety]^−^	Quercetin-*O*-acetyl glucoside	5.72 ± 0.53	3.44 ± 1.01
**20**	13.49 13.52	445	269 [apigenin moiety]^−^	Apigenin 7-*O*-glucuronide	1.20 ± 0.37	9.47 ± 4.09
**21**	13.70 13.74	359	315 [M-H-CO_2_]^−^ 225 [M-H-C_8_H_8_O_2_]^−^ 197 [M-H-caffeoyl]^−^ 181 [M-H-*O*-caffeoyl]^−^ 179 [caffeoyl moiety]^−^	Rosmarinic acid	8.73 ± 0.29	2.18 ± 0.49
**22**	14.37	489	447 [M-H-CH_3_CO]^−^ 285 [luteolin moiety]^−^ 151 ^1,3^A^−^	Luteolin-*O*-acetylglucoside	1.43 ± 0.19	/
**23**	14.89	537	519 [M-H-H_2_O]^−^ 269 [apigenin moiety]^−^	2,8”-Biapigenin	/	0.73 ± 0.09
**24**	17.59 17.64	359	344 [M-H-CH_3_]^−^ 237 [M-H-C_7_H_6_O_2_]^−^ 223 [M-H-C_8_H_8_O_2_]^−^ 221 [M-H-C_8_H_10_O_2_]^−^ 123 [3-methoxyphenyl moiety]^−^	Ophiopogonanone E	NQ	NQ
**25**	19.05	329	314 [M-H-CH_3_]^−^ 301 [quercetin moiety]^−^ 299 [M-H-2CH_3_]^−^	Dimethylquercetin	2.72 ± 0.27	/
**26**	19.53	359	344 [M-H-CH_3_]^−^ 237 [M-H-C_7_H_6_O_2_]^−^ 223 [M-H-C_8_H_8_O_2_]^−^	Ophiopogonanone E isomer	NQ	/
**27**	19.84 19.87	309	291 [M-H-H_2_O]^−^ 265 [M-H-CO_2_]^−^	Oxodihydroxy octadecenoic acid	NQ	NQ
**28**	21.09	329	311 [M-H-H_2_O]^−^ 285 [M-H-CO_2_]^−^	Trihydroxy octadecenoic acid	NQ	/
**29**	21.39	343	328 [M-H-CH_3_]^−^ 196 ^1,3^A^−^	5,2’-Dihydroxy-7,8,6’-trimethoxyflavone	0.71 ± 0.11	/
**30**	21.81	311	293 [M-H-H_2_O]^−^ 267 [M-H-CO_2_]^−^	Dihydroxy octadecadienoic acid	NQ	/
**31**	23.60	339	339, 275, 183	Unknown	NQ	/

Retention time (RT), pseudomolecular ion ([M-H]^−^), ethanol extract (EE), aqueous extract (AE), not quantified (NQ), caffeoylquinic acid (CQA), * adduct with formic acid. The identification of peaks was confirmed by comparing their retention times and MS and DAD spectra with those of standards injected in the same conditions, or by comparing with MS and UV-Vis data previously reported in the literature. For each compound identified, the quantification was carried out using the method of external standards and calibration curves produced with pure standards that were structurally closest to it.

**Table 3 molecules-27-08657-t003:** Antioxidant activity of *S. hispidula* extracts by DPPH^•^, ABTS^●+^, and FRAP assays.

Sample	DPPH^●^	ABTS^●+^	FRAP
	IC_50_ (μg/mL)		A_0.5_
*S. hispidula* EE	5.67 ± 0.07	5.95 ± 0.82	10.77 ± 0.12
*S. hispidula* AE	2.43 ± 0.11	2.16 ± 0.1	5.82 ± 0.17
Ascorbic acid	3.86 ± 0.23	4.01 ± 0.09	10.03 ± 0.22

IC_50_ and A_0.50_ values correspond to the concentration of 50% inhibition percentage and the concentration at 0.5 absorbance, respectively. EE: ethanolic extract; AE: aqueous extract.

**Table 4 molecules-27-08657-t004:** Enzyme inhibitory activities of *S. hispidula* extracts against the activity of α-glucosidase and α-amylase enzymes.

Sample	Enzyme Inhibitory Activity	
	α-Glucosidase IC_50_ (μg/mL)	α-Amylase (% inhibition)
*S. hispidula* EE	106.94 ± 1.55 ^a^	30.34 ± 4.58
*S. hispidula* AE	23.52 ± 6.33 ^b^	4.86 ± 0.04
Acarbose	405.77 ± 34.83 ^c^	nd *

Values followed by different superscript letters are significantly different at *p* < 0.05 (HSD Tukey test). * Because of the very high concentrations required to produce stronger inhibitions, it was not possible to determine the IC_50_ value. EE: ethanolic extract; AE: aqueous extract, nd: not determined.

## Data Availability

Data are contained within the article.

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
