# Peer review of "Metabolite Profiling, Antioxidant and Key Enzymes Linked to Hyperglycemia Inhibitory Activities of Satureja hispidula: An Underexplored Species from Algeria"

_molecules, 2022, doi:10.3390/molecules27248657_

Round 1

Reviewer 1 Report

In this study, the authors explored the extracts from the aerial parts of the endemic species Satureja hispidula by ultra-high-performance liquid chromatography coupled with a diode array detector and an electrospray mass spectrometer (UHPLC-DAD-ESI/MS) method. They found some extracts displayed the inhibitory activities on α-glucosidase and α-amylase. Here are some major concerns:

(1) Please add more information for introduction. Current version is so rough. 

(2) Do you have some structure for these two extracts?

(3) Statistic analysis is also rough.. please add more information. 

(4) Does these two extracts display other activities? 

(5) Please divide the results and discussion. 

Author Response

Response to Reviewer 1 Comments

Point 1: In this study, the authors explored the extracts from the aerial parts of the endemic species Satureja hispidula by ultra-high-performance liquid chromatography coupled with a diode array detector and an electrospray mass spectrometer (UHPLC-DAD-ESI/MS) method. They found some extracts displayed the inhibitory activities on α-glucosidase and α-amylase. Here are some major concerns:

Response 1: We start by thanking the reviewer's comments and suggestions, which were taken into consideration during the preparation of this new version and naturally contributed to its improvement.

Point 2: Please add more information for introduction. Current version is so rough.

Response 2: We tried to be concise in the introduction, but the reviewer is right. We add more information.

Point 3: Do you have some structure for these two extracts?

Response 3: We add the major compounds’ structure as well as their fragmentation pattern.

Point 4: Statistic analysis is also rough. please add more information.

Response 4: We add more information to complete that point.

Point 5: Does these two extracts display other activities?

Response 5: At this moment, we do not know, but mainly considering the antioxidant activity displayed by the extracts, we are searching for collaborations to perform antitumor and anti-inflammatory assays.

Point 6: Please divide the results and discussion.

Response 6: We changed it accordingly to the reviewer's suggestion.

Reviewer 2 Report

In this manuscript, the authors descripted the identification and quantification of the phytochemical constituents of aerial parts of the endemic species S. hispidula growing in Algeria by UHPLC-DAD-ESI/MS method, as well as the assessment of their antioxidant and antidiabetic effects. The findings provide more evidence for the use of S. hispidula in Algerian folk medicine as a treatment of hyperglycemia and the development of new anti-diabetic medications. Overall, the manuscript is good for publication, but I have the following questions:

1.    What is the percentage of formic acid in your LC solvent?

2.    What is the gradient of the LC?

3.    The paper said the retention time from literature is one of the parameters used for compound identification, did you use the same column and gradient so that you can compare the RT?

4.    Can the mass spec setting be descripted in detail?

Author Response

Response to Reviewer 2 Comments

Point 1: In this manuscript, the authors descripted the identification and quantification of the phytochemical constituents of aerial parts of the endemic species S. hispidula growing in Algeria by UHPLC-DAD-ESI/MS method, as well as the assessment of their antioxidant and antidiabetic effects. The findings provide more evidence for the use of S. hispidula in Algerian folk medicine as a treatment of hyperglycemia and the development of new anti-diabetic medications. Overall, the manuscript is good for publication, but I have the following questions:

Response 1: We start by thanking the reviewer's comments and suggestions, which were taken into consideration during the preparation of this new version and naturally contributed to its improvement.

Point 2: What is the percentage of formic acid in your LC solvent?

Response 2: Thank you for calling our attention to the missing information. Now it is indicated that mixture A is 0.1% formic acid in water.

Point 3: What is the gradient of the LC?

Response 3: We tried so much to be concise, and then forgot to include important information. The required information was added.

Point 4: The paper said the retention time from literature is one of the parameters used for compound identification, did you use the same column and gradient so that you can compare the RT?

Response 4:. Actually, what we did was the injection of standard compounds in the same conditions and used their data. We now add that information “The identification of peaks was confirmed by comparing their retention times and MS and DAD spectra with those of standards injected in the same conditions”.

Point 5: Can the mass spec setting be descripted in detail?

Response 5: We add more details concerning mass spectrometry fragmentation.

Round 2

Reviewer 1 Report

I appreciate the authors' effort for this revision. I have no comments now.